# Predictive Value of Cervical Shear Wave Elastography in the Induction of Labor in Late-Term Pregnancy Nulliparous Women: Preliminary Results

**DOI:** 10.3390/diagnostics13101782

**Published:** 2023-05-18

**Authors:** Tatiana Costas, María de la O Rodríguez, Mercedes Sánchez-Barba, Juan Luis Alcázar

**Affiliations:** 1Department of Obstetrics and Gynecology, Complejo Asistencial Universitario de Salamanca, 37001 Salamanca, Spain; mariolarodrimartin@usal.es; 2Group of Investigation in Obstetrics and Gynecology, Biomedical and Diagnostic Sciences Department, University of Salamanca, 37001 Salamanca, Spain; 3Group of Investigation in Cardiovascular and Renal Pathophysiology, Physiology and Pharmacology Department, Biomedical and Diagnostic Sciences Department, University of Salamanca, 37001 Salamanca, Spain; 4Biostatistics Department, University of Salamanca, 37001 Salamanca, Spain; 5Department of Obstetrics and Gynecology, Clínica Universidad de Navarra, 31008 Pamplona, Spain

**Keywords:** shear wave elastography, induction of labor, late-term pregnancy

## Abstract

The prediction of induction of labor continues to be a paradigm nowadays. Bishop Score is the traditional widely spread method but with a low reliability. Ultrasound cervical assessment has been proposed as an instrument of measurement. Shear wave elastography (SWE) should be a promising tool in the prediction of the success of labor induction in nulliparous late-term pregnancies. Ninety-two women with nulliparous late-term pregnancies who were going to be induced were included in the study. A shear wave measurement of the cervix divided into six regions (inner, middle and outer in both cervical lips), cervical length and fetal biometry was performed by blinded investigators prior to routine hand cervical assessment (Bishop Score (BS)) and induction of labor. The primary outcome was success of induction. Sixty-three women achieved labor. Nine women did not, and they underwent a cesarean section due to failure to induce labor. SWE was significantly higher in the inner part of the posterior cervix (*p* < 0.0001). SWE showed an area under the curve (AUC): 0.809 (0.677–0.941) in the inner posterior part. For CL, AUC was 0.816 (0.692–0.984). BS AUC was 0.467 (0.283–0.651). The ICC of inter-observer reproducibility was ≥0.83 in each region of interest (ROI). The cervix elastic gradient seems to be confirmed. The inner part of the posterior cervical lip is the most reliable region to predict induction of labor results in SWE terms. In addition, cervical length seems to be one of the most important procedures in the prediction of induction. Both methods combined could replace the Bishop Score.

## 1. Introduction

Induction of labor (IOL) is a common practice in many obstetrics situations such as prolonged pregnancies. It is one of the most frequent procedures in Obstetrics, and its use increased worldwide from 9.5% to 23.2% between 1990 and 2009 [1,2]. Induction of labor at 41 weeks of gestation is associated with a slight but significant decrease in perinatal mortality, so induction of labor can be offered to women in the interval between 41 and 42 weeks with a degree of recommendation A [3,4]. Induction methods may vary according to guidelines, but are usually, combinations of cervical ripening using mechanical or pharmacological methods followed by the production of contractions with oxytocin and early amniotomy [5,6,7]. It is known that IOL has a large impact on the increase in maternal-fetal well-being, for example, cesarean delivery. For that reason, each IOL should be correctly indicated and studied [8,9].

Many techniques have been studied to assess the uterine cervix before induction of labor to predict its success. The classical and most widely used method to assess cervical status is Bishop Score (BS) regardless of its poor predictive value for induction outcome [10,11,12]. Recent studies are looking for the value of cervical ultrasound assessment as an IOL predictor such as cervical length with moderate reliability [13,14]. Although the measurement of cervical consistency has not been possible until elastography arrival.

There are different forms of elastography. Strain elastography is based on the generation of a strain in a tissue due to human movements and comparing the tissue reaction with adjacent tissue that cervical tissue does not have. It is operator dependent and it generates a qualitative value [15]. The first cervical studies with small cohorts were presented with this technique with moderate results [16,17,18,19,20,21,22,23,24]. Shear wave elastography (SWE) is another technique where ultrasound pulses are generated with two waves: one round trip and another perpendicular to the first (shear wave). The shear wave velocity (v) correlates with tissue stiffness. Stiffness could be estimated with Young’s modulus (E) through velocity: E ≅ 3pv^2^ where p is the density of the tissue that is assumed to be constant. It is a non-operator-dependent technique with quantitative results (kilopascals) and good inter-observer reproducibility [25,26,27]. One study shows SWE as a good predictor of IOL [28].

The objective of this study was to examine the value of shear wave elastography in front of other conventional methods such as cervical length or Bishop Score in the prediction of the success of labor induction in nulliparous late-term pregnancies.

## 2. Materials and Methods

### 2.1. Study Design

A prospective observational study with no intention to treat was conducted between March 2021 and November 2022. Nulliparous women admitted to IOL due to late-term pregnancy were invited to participate.

The inclusion criteria were: nulliparous (understood as first pregnancy or if they have a prior pregnancy, it should be a miscarriage or a cesarean section without any cervical modification (as a scheduled cesarean). All pregnancies with any prior cervical dilatation major than 2 cm were ruled out) singleton pregnancies with vertex presentation, estimated fetal weight in the third trimester between the 10th and 90th percentile, no fetal, placenta or amniotic fluid pathologies and gestational age between 40 + 6 and 41 + 5 weeks in the moment of induction.

### 2.2. Sample Size

Sample-size calculations were realized from scarce previous literature. Lu et al. [28] presented a nulliparous subpopulation in their study with an SWE + cervical length (CL) specificity to predict the failure labor induction of 84.2% opposite to the 67.2% specificity of the Bishop Score. The sample size was calculated using specificity and not sensibility due to the importance of prediction without false positives. With these data, the sample size was *n* = 87 in order to detect a 17% of specificity difference between both methods. Statistical power was established in 80% and an alpha error of 5% and 10% of sample loss was assumed.

### 2.3. Cervical Shear Wave Elastography Measurement

During the IOL schedule, participants were assessed to participate in the study. A maximum of 24 h prior to IOL, SWE (and other ultrasound assessments) were performed. Women were asked to empty the bladder and in a modified lithotomy, a transvaginal scan of the cervix was performed using the vaginal probe PVT-781VTE of the Canon Aplio i700 Ultrasound Machine (Canon Medical Systems Corporation, Otawara, Japan). The midsagittal view of the cervix was identified by aligning the probe with the cervical canal. Under Fetal Medicine Foundation criteria were used for ultrasound cervical assessment; the ultrasound image was optimized with a 2/3 part of image magnification of the cervix performed in the first moment the CL measure in the B-mode [29] and on repeat we followed the SWE procedure.

Shear wave assessment was performed by gently introducing the probe with minimum tissue pressure with transducer placement at the anterior fornix. The probe was withdrawn to minimize transducer pressure on the cervix while not compromising the B-mode image [30,31]. The excess gel was removed.

The sample SWE box was optimized, taking into account elastography and propagation map (less propagation distortion, more planar, equidistant and perpendicular waves, best color map capture) between 20 mm in the anteroposterior dimension and 20 mm across the width measured at the deepest part of the elastogram [31]. That consideration is because the recent European Federation for Ultrasound in Medicine and Biology update on the use of liver ultrasound elastography recommends that the main pulse focus should be placed at the level of the ROI [32]. The elastogram was stable for at least 3 s before kilopascal measurements were obtained [33].

The ARFI pulse frequency used was 4 MHz. The range of kilopascals was adjusted between 0 kilopascals and 40 kilopascals in order to differentiate glands. Blue indicates softer tissue. Regions without color were rejected due to non-planar shear wave propagation. Maximum time smoothing was used in order to improve the image and elastographic opacity was set to 0.6. Stabilization time was at least 3 s. Images of technical considerations are shown in Figure 1, Figure 2 and Figure 3.

The cervix was divided into six sub-regions. Three in the anterior lip (inner, middle and outer) and the same in the posterior lip according to the literature [28]. Other studies [31,34,35] use only inner and outer. A 5 mm circle region has been utilized for this study to facilitate ROI placement in the circumferential layer of collagen and smooth muscle of the cervix avoiding the endocervical canal [28,31]. Each part of the cervix was examined separately so that the region of greatest sensitivity could be positioned more effectively. Regions of interest (ROIs) emplacements are shown in Figure 4.

The SWE measurement was repeated twice by two blinded investigators. The average of these two measurements was used for the analysis. Trained sonographers in SWE performed the ultrasound examinations. The inter-observer reproducibility was assessed in the 40 first women, blinded to each other. 

In addition, a modified biophysical profile was realized. An estimated fetal weight (EFW) (Hadlock formula estimated by biometry), placenta grade and maxim column amniotic fluid were also measured.

### 2.4. Induction of Labor

Induction of labor procedure was conducted in the next 24 h to SWE. A blinded obstetrician calculated the Bishop Score (BS) by performing a vaginal digital examination. 

The induction methods depend on clinical history and BS. If BS was equal to or more than 6 points, an induction with oxytocin infusion or amniotomy was carried out. If BS was less than 6 points, the ripening cervical method depends on clinical records. If the woman had had a previous cesarean or uterine surgery or she presented a high risk of hyperstimulation, Cook Cervical Ripening Balloon should be used for 12 h. If gestation was low risk, PG E1 prostaglandin could be administrated (vaginal administration of misoprostol 25 µg tablets (Misofar, Pfizer, New York, NY, USA) in a 6 h interval with a maximum of four administrations. For doubtful cases, we may administrate PGE2 (dinoprostone pessary 10 mg) (Propess, Ferring, Kiel, Germany) for 12 h [36]. 

### 2.5. Failure to Progress, Cesarean Indications

Successful induction of labor (SIOL) was defined as the accomplishment of cervical effacement with at least 4 cm dilatation. For that reason, failure to enter the active phase (FIOL) was defined as a lack of dilatation of more than 4 cm after amniotomy and 18 h of oxytocin infusion—which causes at least four to five contractions every ten minutes—according to national guidelines [37].

Other emergency cesarean indications, apart from failure to enter the active phase (FIOL), were as follows: failure to progress (FTP) (no dilatation advance in 4 h over 1 cm), risk of loss fetal well-being (RLFB) (presence of pathological reassuring non-stress fetal test which requires delivery immediately), cephalo-pelvic disproportion (CPD) (when the pelvis is too narrow to allow the passage of the fetus) or any other condition which requires prompt delivery [37].

### 2.6. Statistical Analyses

The primary outcome was successful induction of labor (SIOL). RLPB cesarean prior to achieving SIOL was excluded since the decision to perform a cesarean section is completely irrelevant to cervical status. 

Maternal and fetal characteristics, ultrasound findings and shear wave parameters were compared in each group. The normality of variables was tested using the Kolmogorov–Smirnov test or Shapiro–Wilk test. Normally distributed continuous variables were compared using Student’s *t*-test and non-normal ones with the Mann–Whitney U test. Inter-observer reproducibility was assessed by the intra-class correlation coefficient (ICC). Wilcoxon signed rank test was used to compare SWE values between different regions. BIPLOT observational graphics were used to assess the combined distribution between SWE six measurements variables and the outcomes.

Point bi-serial correlation was used to determine the correlation between SWE measurements and outcomes. Logistic univariate regression was used to determine independent prediction factors. Multiple logistic regression was attempted with significant variables in univariate analysis. 

In order to assess the discriminatory power to detect FIOL for each diagnostics method, the receiver operating characteristics curve (ROC) was calculated. Additionally, the area under the curve (AUC) and confidence intervals. 

Data were analyzed using SPSS version 29.0 (IBM Corporation NY, USA. License under University of Salamanca). A two-tailed *p*-value < 0.05 was considered statistically significant. A BIPLOT graphic was made with Jamovi version 2.32.22 (AGPL3 license, Sydney, Australia).

## 3. Results

### 3.1. Population Characteristics

During the study period, 92 women were recruited (94 women met the inclusion criteria but two declined to participate). The first four were excluded due to technique enhancements. In 14 of them, a cesarean section prior to achieving 4 cm was necessary due to RLPB. One of them was excluded from the analysis phase owing to not fulfilling FIOL cesarean indications at the moment of the cesarean section (Figure 5). 

Demographic characteristics of the women that participated in the study are reported in Table 1. All of them were nulliparous women induced due to late-term pregnancy without any pathology in their gestation.

The induction method was as follows: in nine cases (12.3%) directly with oxytocin due to Bishop ≥ 6, in 49 (67.1%) women, dinoprostone was administrated. A Cook Balloon was used in seven (9.6%) of the inductions and misoprostol in eight (11%) of the women. The medium BS pre-induction was 2.6 (0–10). After cervical ripening (if needed) medium BS was 5.52 (0–12). The average time of total induction was 22.7 (5–48) h. The cervical ripening mean was 12.1 (1–27) h. Hours of oxytocin infusion was 8.79 h (no needed—24 h). Maximum oxytocin level was 27.81 mL/h (no needed—102 mL/h).

There was only a case of pathological reassuring non-stress fetal test during cervical ripening. In 19 (26%) the cardiotocographic register was pathological or not appeasing during oxytocin administration and extra control or speeding up of labor was necessary. The artificial rupture of membranes mean hours was 14.47 (0–30) h. Meconium was present in 16 (22%) of the partum. Epidural was used in 66 (90.4%) of the cases. The average time since starting induction and epidural administration was 13.26 (2–33) h. 

Sixty-four (87.7%) women achieved successful induction (SIOL). Vaginal delivery was possible in 55 (75.4%) of the cases, 29 of them (39.7%) were instrumental deliveries. A cesarean section was performed in 18 (24.6%) of the women. Four women (22.2%) were FTP cesarean, nine patients (50%) were FIOL cesarean, two (11.1%) of them were due to CPD and three (16.7%) cases, cesarean section was due to RLFB, also 14 were excluded prior to analysis. A comparison between FIOL and SIOL group maternal characteristics, ultrasound b mode findings, SWE measurements and BS and birthweight are shown in Table 2.

### 3.2. SWE Measurements 

The ICC of inter-observer reproducibility was ≥0.83 in each ROI (M1 ICC 0.94, M2 ICC 0.90, M3 ICC 0.83, M4 ICC O.92, M5 ICC 0,89, M6 IC 0.88).

The cervix elastic gradient seems to be confirmed. The inner part of the cervix appears to be harder than the middle and middle to the outer part in both cervical lips (8.40 Kpa > 7.75 Kpa > 7.45 Kpa in the anterior lip and 9.60 kpa > 8.65 Kpa > 8.40 Kpa in posterior lip). Data are shown in Table 3. 

BIPLOT observational graphics were used to assess the combined distribution between SWE six measurement variables and FIOL/SIOL results. As the graph shows, there is an intrinsic relation between M1, M2 and M3 and M4, M5 and M6 but not between SWE measurements in the anterior and posterior lips. In addition, M4, M5 and M6 show a better correlation with FIOL than SWE measurements in the anterior lip. Furthermore, M4 presents less variability than M5 and M6 (Figure 6).

The correlation between shear wave measurements and the results of induction were shown in Table 4. Only SWE in the inner posterior lip (M4), inner mean SWE and posterior mean SWE have statistical significance. M4 SWE is part of both other measurements and presents a higher significant correlation.

### 3.3. ROC Curves 

ROC curves were analyzed to explore the efficacy of each measurement method (Bishop Score shear wave elastography (SWE) and cervical length) in the prediction of the result of induction. For cervical length, AUC was 0.816 (0.692–0.984). Bishop Score (BS) AUC was 0.467 (0.283–0.651), including a 0.5 value. For SWE M4 (inner part of the posterior lip) AUC was 0.809 (0.677–0.941) (Figure 7). Sensitivity, specificity, positive predictive value, negative predictive value and likelihood ratio are shown in Table 5.

### 3.4. Univariate and Multivariate Analysis 

Table 6 shows the odds ratio (OR) in univariate analysis to predict the result of induction. The results indicate that independent predictors are fetal percentile prior induction, HC (fetal head circumference), AC (fetal abdomen circumference), Shear wave measurement in the inner posterior cervical part (M4) and shear wave mean of the posterior lip.

Table 7 shows the multivariate analysis for the prediction of the result of induction. Adjusted odds ratio (AOR) was only significant for shear wave measurement in the inner posterior cervical part (M4). AC and HC were not included due to their collinearity with the fetal percentile prior to delivery. SWE mean posterior lip was not included for the same reason.

### 3.5. Correlation with Hours of Delivery

The mean hours from induction starting to delivery (vaginal delivery or cesarean) was 21.77 h (13.75–29.56) in the SIOL group and 20.56 h (25.50–32.50) in the FIOL group. There are statistical differences between them (*p* < 0.001).

The hours of induction have a weak but significant correlation with SWE M4 (correlation of 0.297 *p* 0.011).

## 4. Discussion

There is not a wide range of studies about SWE and the prediction of induction of labor (IOL). As far as we know there are only two studies that use SWE [28,34]. Others had explored SWE in obstetrics cervix but not for IOL prediction [26,27,35,38].

First of all, the intra and inter-observer reproducibility of the technique has been demonstrated. Our results are similar to Lu et al. (ICC at least >0.85 in each ROI) [28], Duan et al. (ICC at least >0.87 in each ROI) [27] and Peralta et al. (ICC at least >0.89 in each ROI) [26]. Torres et al. present ICC 0.18 for the internal cervix area; it may be due to the ROI position in the middle of the cervical canal and the SWE technique in contrast to other studies [34].

The decreasing stiffness from the inner to the outer part in both cervical lips is similar in our study to the previous literature [26,27,28,39,40,41]. In our study, it is near to significance maybe due to sample size. Additionally, the anterior cervical lip seems to be less stiff than the posterior (lip) [26,27] in contrast to Lu et al. [28]. This comparison is not possible in all studies due to different ROI positions. In addition, the absolute value is different between Lu et al. [28] and us presenting their study lower SWE values in all ROI. It might be due to the specificity in our sample (later term nulliparous pregnancies without any cervical ripening). Our values (FIOL group M1: 10.46 Kpa, M4: 13.91 Kpa) are similar to the 28–32 + 6 weeks nulliparous group that presents Duan et al. (“M1”: 11.7 Kpa, “M4”: 12.5 Kpa) [27].

This differential in stiffness could be explained due to internal cervical composition. Collagen fiber orientation and muscle composition are different in each region. The cervical canal is surrounded by a layer of longitudinal smooth muscle fibers adjacent to the canal. Wrapping circumferentially around the longitudinal layer is a layer of smooth muscle and collagenous cells. The muscle concentration is decreasing from the inner to the outer part [42,43]. In addition, the collagen crosslinks are different in each part. The heterogeneity in the inner part is higher than in the external one. Moreover, the inner posterior cervical part has less collagen content and a higher proportion of deoxypyridinoline (DPD) and pyridinoline (PYD) than the inner anterior cervical part [44].

Torres et al. present an AUC for the prediction of SIOL to SWE in the internal cervix of 0.652 and 0.689 for the external cervix. Global SWE AUC was 0.672 (ROI position is situated near the middle of the canal) [34]. In a nulliparous subpopulation of their study, Lu et al. show an AUC combined for cervical length and SWE in the internal cervix of 0.816 (0.888 for all parous women) [28] close to our AUC for the inner posterior part (M4) 0.809.

In a meta-analysis reported by Londero et al. in 2016, four studies in which strain was used to predict IOL were included [18,19,20,21] with a pool-estimated sensitivity of 71.1% and specificity of 54.7% to IOL [17]. SWE studies especially improve specificity. Wozniak et al. describe a significant difference between strain value in the elastographic index in the inner cervical canal between successful and unsuccessful labor [22]. Zhou et al. present an AUC for strain in the inner part of the cervix to predict vaginal delivery of 0.645 [45]. The elastography strain index was also significantly higher in the inner part of the cervix in the FIOL group presented by Swiatkowska-Freund et al. [16]. Other strain studies such as Hamza et al. [24] and Pereira et al. [23] did not find any significance using strain elastography.

The inner part of the cervix seems to be the most reliable region to predict IOL in terms of elastography. Only our study presents and works with data from both cervical lips separately. Due to our data and the internal cervix composition (the inner part of the cervix is the star point of cervical maturation with a great muscle proportion and increased heterogeneity of crosslink proteins, there are differences in the posterior inner part compared to the anterior inner part, with an uncertain significance in term of maturation) [44], the posterior inner part of the cervix appears to be a key point in IOL prediction.

In addition, the ROI position is an intricate point in SWE. Lu et al. [28] and we both placed the ROI in the circumferential layer of collagen and smooth muscle of the cervix avoiding the endocervical part and canal. Torres et al. inserted a ROI in the middle of the canal (most heterogeneous and anisotropic zone). Tissue anisotropy and high acoustic attenuation in the cervix have been discussed [46,47,48]. With an SWE box optimized by taking into account the elastography and propagation map together with a correct ROI position in the circumferential layer, Young’s modulus can be calculated with reliability [31]. It has also been demonstrated that depth does not affect SWE [40].

The major strengths of our study are the homogenization of the sample and the establishment of a rigorous based-in-science method of SWE measurement procedure. In this way, we demonstrated that the inner part of the posterior cervical lip is the most reliable region to predict the results of labor induction in SWE terms. The cervical length also seems to be one of the most important procedures in the prediction of induction. Both methods combined could replace the widely spread Bishop Score.

## 5. Drawbacks

Although the shear wave elastography curve seems to be easy, a minimum of experience with the technique is necessary to apply it. Our simple size with only nine FIOL cesarean sections is the main weak point. Thus, we are unable to create a multivariable analysis. Additionally, correlations between induction methods and successful induction are not possible due to the same reason. Further multicenter studies with rigorous measurement protocols should be developed to strengthen the value of SWE in the prediction of IOL.

## Figures and Tables

**Figure 1 diagnostics-13-01782-f001:**
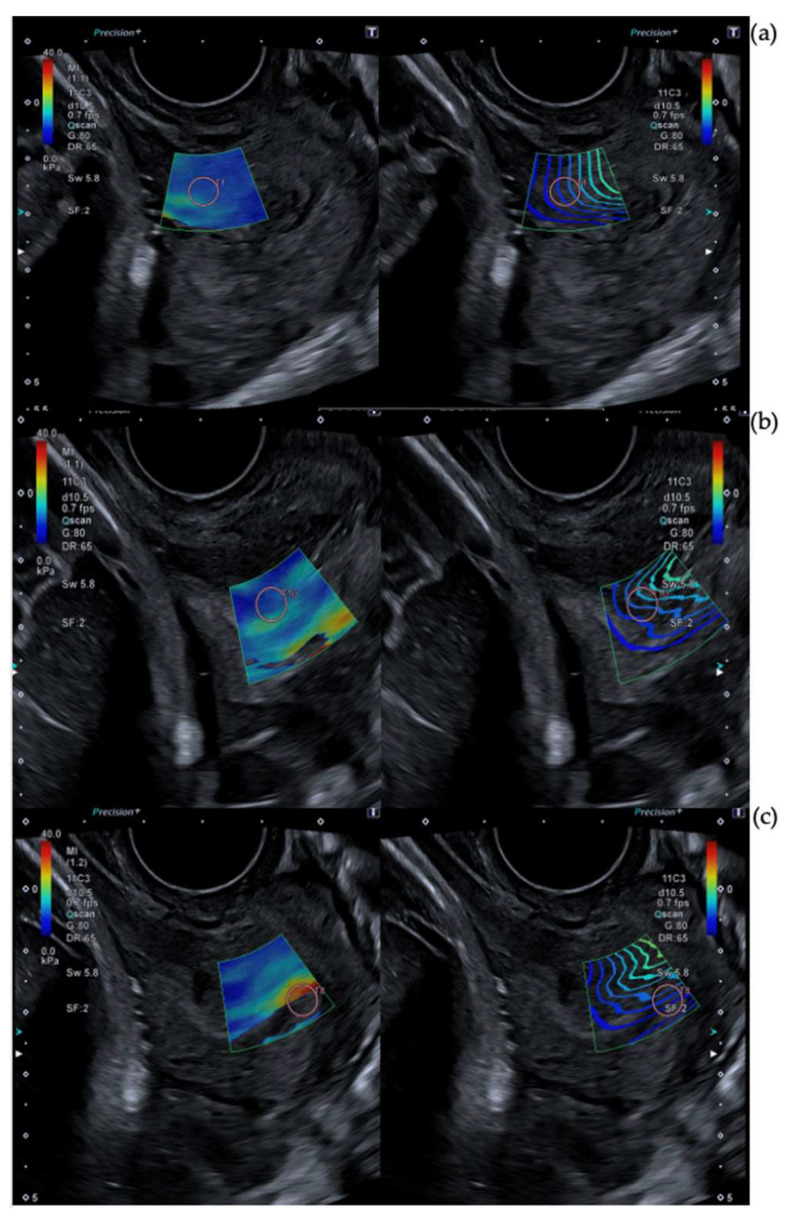
Technical considerations of SWE measurements. Figure 1 shows different SWE captures. The left image is a color map of elastography and the right one is a propagation map. Both of them overlay b-mode ultrasound images. (**a**) Represents a correct ROI position with an adjusted elastography box and linear waves in the propagation map in the inner anterior lip. (**b**) Shows a wrong capture of SWE although the color map is complete and the linear map is more or less linear, ROI emplacement is incorrect due to its position in the middle of the canal. (**c**) Represents a wrong capture due to ROI emplacement, no color caption, and high distortion in the propagation map.

**Figure 2 diagnostics-13-01782-f002:**
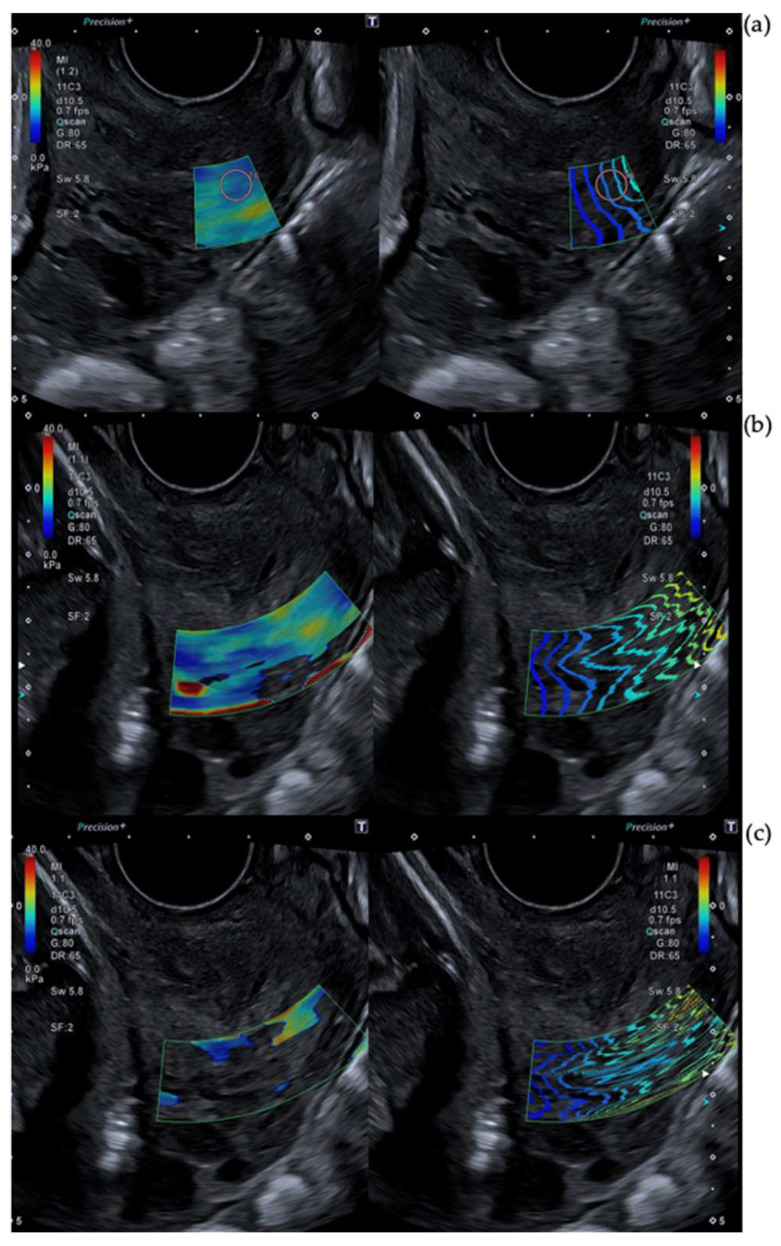
Technical considerations of SWE measurements: Box size. Figure 2 shows different SWE captures. The left image is a color map of elastography and the right one is a propagation map. Both of them overlay b-mode ultrasound images. (**a**) shows a right caption in the outer posterior lip. (**b**,**c**) exhibit color map not complete and strong distortion in propagation map. A large caption box creates the issue. Not reliable measurements should be obtained there.

**Figure 3 diagnostics-13-01782-f003:**
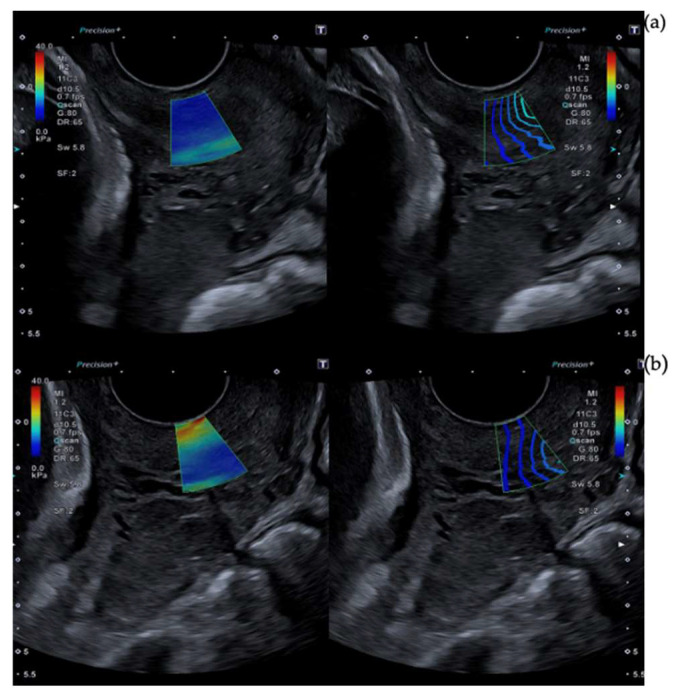
Reflection of excess pressure in SWE measurements. Figure 3 shows different SWE captures. The left image is an elastography color map, and the right one is a propagation map. Both of them overlay b-mode ultrasound images. (**a**) Represents a capture without any tissue pressure. (**b**) is the same image but with an excess of pressure that deforms the b-mode. This generates a false stiffness.

**Figure 4 diagnostics-13-01782-f004:**
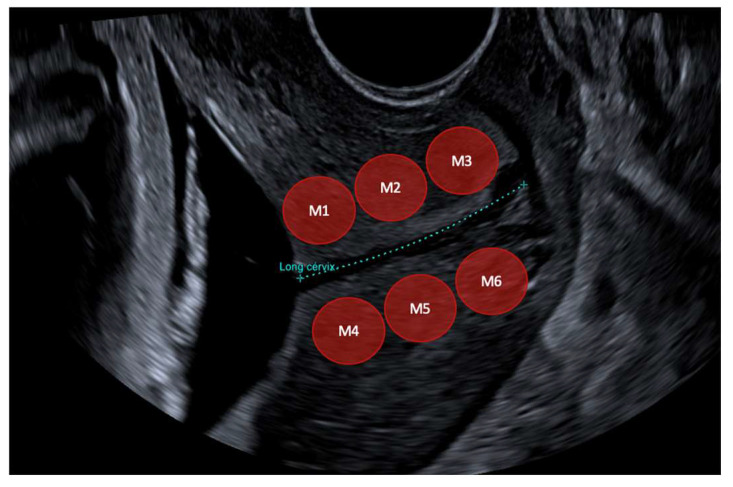
Figure 4 shows ROIs positions. Cervix was divided into six sub-regions. Three in the anterior lip (M1—inner, M2—middle and M3—outer) and the same in the posterior lip (M4—inner, M5—middle and M6—outer). A 5 mm circle region has been utilized to facilitate ROI placement in the circumferential layer of collagen and smooth muscle of the cervix avoiding the endocervical canal.

**Figure 5 diagnostics-13-01782-f005:**
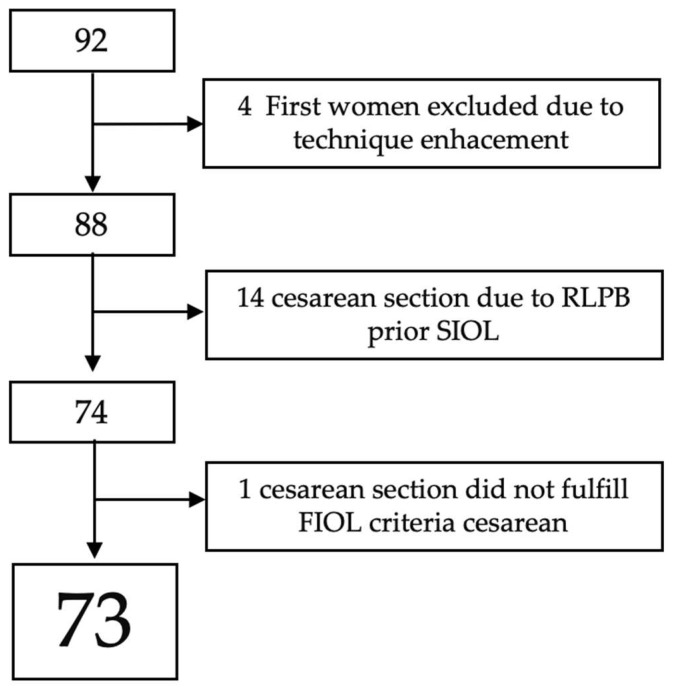
Figure 5 shows the flowchart of patients’ selection with missing patients.

**Figure 6 diagnostics-13-01782-f006:**
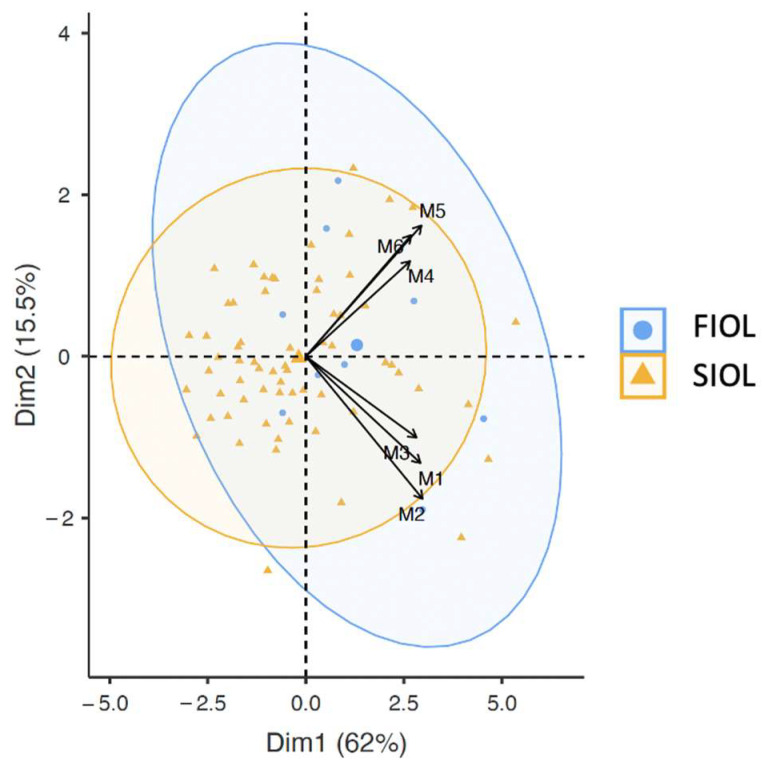
BIPLOT observational graphics between shear wave measurements and FIOL/SIOL. On the one hand M1, M2 and M3 and on the other hand M4, M5 and M6 measurements are related due to their angles being slight. No relation between both groups due to the straight angle between them. In addition, M4, M5 and M6 show a better correlation with FIOL (blue spots are mostly in their quadrant). M4 and M3 present less variability (less length of arrow).

**Figure 7 diagnostics-13-01782-f007:**
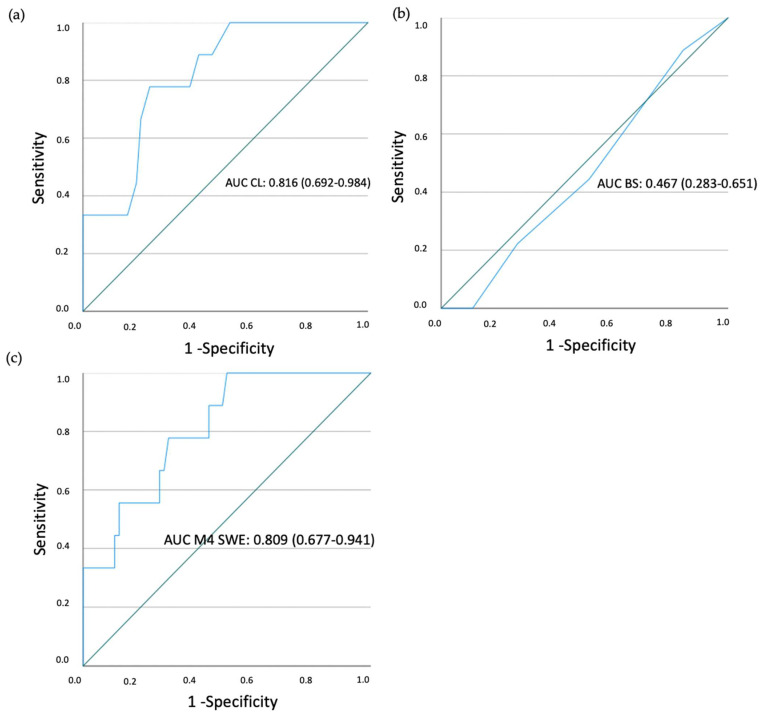
ROC curves. For cervical length, AUC was 0.816 (0.692–0.984) (**a**). BS AUC was 0.467 (0.283–0.651) including 0.5 value (**b**). For SWE M4 (inner part of the posterior lip) AUC was 0.809 (0.677–0.941) (**c**). Abbreviation: ROC—receiver operating characteristics curve, AUC—area under curve, CL—cervical length, SWE—shear wave elastography.

**Table 1 diagnostics-13-01782-t001:** Demographic characteristics of women included in the study.

Characteristics	Value
Maternal age (y)	33 (18–44)
Maternal height (cm)	164 (150–175)
BMI before pregnancy (kg/m^2^)	24.13 (16.5–43.3)
BMI at delivery (kg/m^2^)	28.82 (20.8–44.4)
Smokers	7 (15.8%)
Reproduction techniques	14 (19.2%) (In vitro 13, artificial insemination 1)
Cervical surgery or manipulation	10 (13.7%) (9 curettages)
Prior cesarean section	8 (10.9%)
Gestational age induction (wk)	41.3 (41.0–41.5)
EFW in 3rd trimester (gr)	2589 (2079–3253)
Birthweight (gr)	3461 (2795–4405)
Newborn pH	7.2 (6.47–7.36)

The value is given as median (range) or n (%). Abbreviations: BMI—body mass index, EFW—estimated fetal weight. Measuring units are expressed in parentheses.

**Table 2 diagnostics-13-01782-t002:** Comparison between FIOL and SIOL group maternal characteristics, ultrasound b mode findings, SWE measurements and BS and birthweight are shown.

Characteristics	SIOL (*n* = 64)	FIOL (*n* = 9)	*p*
Maternal age (y)	33 (30.20–36.0)	31.8 (29.0–36.0)	0.602
Maternal height (cm)	164 (161–170)	163 (160–166)	0.432
BMI before pregnancy (kg/m^2^)	23.95 (21.51–25.76)	25.42 (21.03–27.21)	0.657
BMI at delivery (kg/m^2^)	28.53 (25.56–31.12)	30.90 (26.38–32.92)	0.257
EFW in 3rd trimester (gr)	2578 (2411–1752)	2667 (2442–2893)	0.319
Fetal percentile 3rd trimester	40.88 (27.0–51.0)	52.89 (37.50–68.50)	0.077
EFW prior delivery (gr)	3573 (3295–3602)	4033 (3683–4335)	0.030
Fetal percentile prior delivery	33.0 (14.0–44.0)	61.0 (37.50–83.50)	0.020
BPD (mm)	94.11 (91.0–98.0)	96.56 (92.20–101.0)	0.152
HC (mm)	336.31 (326.25–347.0)	348.0 (335.0–358.0)	0.030
AC (mm)	347.73 (332.50–362.75)	365.33 (349.0–384.50)	0.010
Bishop Score	2.73 (1.0–3.75)	2.11 (1.0–3.0)	0.478
Cervical length (mm)	25.12 (20.0–32.0)	42.33 (30.5–49.0)	0.020
SWE M1 (Kpa)	8.86 (6.63–9.98)	10.46 (7.75–13.77)	0.347
SWE M2 (Kpa)	7.94 (6.21–8.73)	9.72 (7.37–12.35)	0.062
SWE M3 (Kpa)	7.45 (5.87–8.61)	8.31 (6.65–9.82)	0.273
SWE Mean anterior lip (Kpa)	8.08 (6.34–8.87)	9.49 (7.10–11.94)	0.113
SWE M4 (Kpa)	9.13 (6.98–11.16)	13.91 (10.0–17.900)	0.023
SWE M5 (Kpa)	8.92 (6.71–10.88)	10.60 (9.05–12.25)	0.052
SWE M6 (Kpa)	8.15 (6.38–9.86)	8.75 (7.57–9.95)	0.288
SWE Mean posterior lip (Kpa)	8.73 (6.86–10.49)	11.09 (9.05–13.76)	0.022
Mean SWE inner	9.01 (7.41–10.58)	12.18 (9.16–15.18)	0.071
Mean SWE outer	7.81 (6.41–8.69)	8.53 (7.83–9.30)	0.090
Birthweight (gr)	3443 (3211–3718)	3590 (3160–3967)	0.292
Total hours of induction (h)	21.77 (13.75–28.0)	29.56 (25.50–32.50)	<0.001

Data are given as median (interquartile range). Abbreviations: BMI—body mass index, EFW—estimated fetal weight. BPD—biparietal diameter, HC—head circumference, AC—abdominal circumference; SWE—shear wave elastography. Measuring units are expressed between parentheses. Mann–Whitney U test was used in maternal age, BMI categories, BPD, Bishop Score, Cervical length, SWE M1, SWE M3, SWE mean anterior lip, SWE inner and SWE outer. Rest of categories are analyzed by Student’s *t*-test.

**Table 3 diagnostics-13-01782-t003:** Mean of Shear Wave elastography in each ROI.

Cervical Region	Inner	Middle	Outer	Inner vs. Middle	Middle vs. Outer
Anterior cervical lip	8.40 (7.15–10.30)	7.75 (6.25–9.40)	7.45 (5.95–8.85)	*p* < 0.01	*p* 0.06
Posterior cervical lip	9.60 (7.40–11.85)	8.65 (7.0–11.20)	8.40 (6.65–10.10)	*p* 0.06	*p* 0.01

Data are given as median (interquartile range).

**Table 4 diagnostics-13-01782-t004:** Correlation between shear wave measurements and FIOL.

Variables	Correlation	*p* Value
M1 SWE	0.166	0.160
M2 SWE	0.215	0.067
M3 SWE	0.128	0.276
M4 SWE	0.437	0.0001
M5 SWE	0.162	0.168
M6 SWE	0.088	0.457
Mean SWE anterior lip	0.192	0.1024
Mean SWE posterior lip	0.289	0.0130
Mean SWE inner	0.369	0.0012
Mean SWE outer	0.131	0.2701

Abbreviations: SWE—shear wave elastography. SWE anterior lip: mean of M1 + M2 + M3. SWE posterior lip mean of M4 + M5 + M6. SWE inner: mean of M1 + M4. SWE outer: M3 + M6.

**Table 5 diagnostics-13-01782-t005:** Predictive value of each diagnostic method of enter in active labor.

Measurement Method	AUC (95% IC)	Cutoff Value	Sensitivity	Specificity	PPV	NPV	LR+	LR−
Bishop	0.467 (0.283–0.651)	>1.5	44%	48%	11%	86%	0.84	1.16
Cervical length	0.816 (0.692–0.984)	>32.5	78%	77%	32%	96%	3.39	0.28
M4 SWE	0.809 (0.677–0.941)	>10.72	78%	70%	27%	96%	2.6	0.31

Abbreviations: AUC—area under the curve, SWE—shear wave elastography, PPV—predictive positive value, NPV—negative positive value, LR+—positive likelihood ratio, LR−—negative likelihood ratio.

**Table 6 diagnostics-13-01782-t006:** Univariate analysis for prediction of result of induction.

	Univariate Analysis		
Variables	Odds Ratio	95% IC	*p* Value
Maternal age	1.06	(0.933; 1.203)	0.372
Maternal height	135.71	(0.01; 2.032)	0.428
BMI before pregnancy	0.937	(0.818; 1.107)	0.352
BMI at delivery	0.905	(0.788; 1.038)	0.154
EFW in 3rd trimester	0.999	(0.996; 1.001)	0.316
Fetal percentile 3rd trimester	0.969	(0.935; 1.004)	0.085
EFW prior delivery	0.997	(0.995; 0.999)	0.07
Fetal percentile prior delivery	0.959	(0.959; 0.932)	0.005
BPD	0.872	(0.732; 1.038)	0.124
HC	0.947	(0.905; 0.992)	0.020
AC	0.939	(0.888; 0.993)	0.028
SME M1	0.875	(0.724; 1.057)	0.166
SWE M2	0.821	(0.821; 0.659)	0.079
SWEM3	0.840	(0.615; 1.147)	0.272
SWE M4	0.701	(0.553; 0.890)	0.04
SWE M5	0.877	(0.726; 1.059)	0.172
SWE M6	0.889	(0.654; 1.207)	0.449
SWE Mean anterior lip	0.810	(0.626; 1.048)	0.109
SWE Mean posterior lip	0.724	(0.552; 0.950)	0.02
Mean SWE inner	0.693	(0.536; 0.896)	0.05
Mean SWE outer	0.814	(0.566; 1.170)	0.267
Cervical Length	0.867	(0.782; 0.962)	0.07
Bishop Score	1.151	(0.642; 2.063)	0.638

Abbreviations: BMI—body mass index, EFW—estimated fetal weight, BPD—biparietal diameter, HC—head circumference, AC—abdominal circumference, SWE—shear wave elastography.

**Table 7 diagnostics-13-01782-t007:** Multivariate analysis for prediction of result of induction.

	Multivariate Analysis		
Variables	Adjusted Odds Ratio	95% IC	*p* Value
SWE M4	1.174	(1.055–1.305)	0.03
Fetal percentile prior delivery	0.997	(0.975–1.020)	0.803

Abbreviations: SWE—shear wave elastography.

## Data Availability

The datasets generated and analyzed during the current study are not publicly available due to local restriction agreements but are available from the corresponding author upon reasonable request.

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
