# Peer review of "Predictive Value of Cervical Shear Wave Elastography in the Induction of Labor in Late-Term Pregnancy Nulliparous Women: Preliminary Results"

_diagnostics, 2023, doi:10.3390/diagnostics13101782_

Round 1

Reviewer 1 Report

I appreciate the opportunity to review the manuscript entitled “Predictive value of cervical shear wave elastography in induction of labor in late-term pregnancy nulliparous women: preliminary results.” submitted to Diagnostics.

Considering the importance of labor induction in the era of growing rates of cesarean deliveries and high percentage of the labor induction failure the authors conducted prospective study to investigate shear wave elastography (SWE) of the uterine cervix as a tool for prediction the success of induction of labor (IOL). SWE was investigated in nulliparous late-term pregnancies.

Reviewer Comments:

1.     Please, clarify lines 73-74 in terms of cervical dilatation.

2.     Authors should explain why the data which always have normal distribution are presented as median and range or IQR. Were there so many outliers that the distribution was not normal? If so, why?

3.     Figure 1. is too condensed. It would be nice if the authors made it clearer for the readers.

4.     Please mention briefly, the national guideline of the labor induction (ref #37) indications and contraindications, and drugs used.

5.     Data on total time needed for vaginal delivery should be added in results (from the beginning of induction to childbirth) and corresponding SWE measurements.

6.     Is it possible to derive a single score for prediction of the success rate of the proposed method?

7.     I suggest the authors add a comment in discussion about influence of various labor induction protocols used in relation to SWE success results. Which was the most frequent method of induction throughout the study?

8.     It would be necessary to mention if all the pregnancies included in the study were physiological post-term pregnancies or there were some pathological conditions (i.e., hypertension, gestational diabetes, etc.)

9.     It would be nice to explain in more detail the rationale of SWE use and mechanisms involved in labor induction success, particularly molecular mechanisms.

10.  Parts of the text contain so many acronyms that it is difficult for readers to follow up the text, please, try to correct it.

11.  Please, add info about drawbacks of the study.

Considering that manuscript provides new and promising data about predictors of successful labor induction with possible clinical usefulness, this submission can be published in Diagnostics following major revision. 

None.

Author Response

Please, see uploaded document

Reviewer 2 Report

The authors present a manuscript which aims to evaluate the predictive value of shear wave elastography in labor induction for nulliparous women with late term pregnancy. The authors have designed and conducted the study properly but unfortunately the manuscript has been written poorly. As the manuscript contains many typographical and grammatical errors, it reuires extensive editing by a professional in English language. Second, the time needed to gain experience for performing shear wave elastography correctly should be addressed as a power-limiting factor for this study. Third, the authors should discuss about the clinical implications of their findings in a separate paragraph of the discussion part. Lastly, all references that were published before 2008 should be replaced with newer and more up-to-date ones if possible. I would recommend that the manuscript can be accepted for publication in Diagnostics after required corrections have been fulfilled.

The authors present a manuscript which aims to evaluate the predictive value of shear wave elastography in labor induction for nulliparous women with late term pregnancy. The authors have designed and conducted the study properly but unfortunately the manuscript has been written poorly. As the manuscript contains many typographical and grammatical errors, it reuires extensive editing by a professional in English language. Second, the time needed to gain experience for performing shear wave elastography correctly should be addressed as a power-limiting factor for this study. Third, the authors should discuss about the clinical implications of their findings in a separate paragraph of the discussion part. Lastly, all references that were published before 2008 should be replaced with newer and more up-to-date ones if possible. I would recommend that the manuscript can be accepted for publication in Diagnostics after required corrections have been fulfilled.

Author Response

Please, see uploaded document

Reviewer 3 Report

The authors reported a very interesting paper examining the value of shear wave elastography in front of other conventional methods such as cervical length or Bishop Score in the prediction of the success of labor induction in nulliparous late-term pregnancies. I have only minor changes to propose, and they should improve the manuscript as detailed below:

- The text should be revised by a native speaker to remove several typos.

- The discussion should be improved by citing relevant and novel key articles about the topic (considering that some references are a bit dated)

Minor editing of English language required. The text should be revised by a native speaker to remove some typos.

Author Response

Please, see uploaded document

Round 2

Reviewer 1 Report

The manuscript has been sufficiently improved to warrant publication in Diagnostics. 

None.